# Neurogenic Inflammation: The Participant in Migraine and Recent Advancements in Translational Research

**DOI:** 10.3390/biomedicines10010076

**Published:** 2021-12-30

**Authors:** Eleonóra Spekker, Masaru Tanaka, Ágnes Szabó, László Vécsei

**Affiliations:** 1Neuroscience Research Group, Hungarian Academy of Sciences, University of Szeged (MTA-SZTE), H-6725 Szeged, Hungary; spekker.eleonora@med.u-szeged.hu (E.S.); tanaka.masaru.1@med.u-szeged.hu (M.T.); 2Interdisciplinary Excellence Centre, Department of Neurology, Albert Szent-Györgyi Medical School, University of Szeged, H-6725 Szeged, Hungary; szabo.agnes.4@med.u-szeged.hu

**Keywords:** primary headache, migraine, trigeminal system, neuropeptides, neurogenic inflammation, animal model, inflammatory soup, dura mater, immune system, migraine treatment

## Abstract

Migraine is a primary headache disorder characterized by a unilateral, throbbing, pulsing headache, which lasts for hours to days, and the pain can interfere with daily activities. It exhibits various symptoms, such as nausea, vomiting, sensitivity to light, sound, and odors, and physical activity consistently contributes to worsening pain. Despite the intensive research, little is still known about the pathomechanism of migraine. It is widely accepted that migraine involves activation and sensitization of the trigeminovascular system. It leads to the release of several pro-inflammatory neuropeptides and neurotransmitters and causes a cascade of inflammatory tissue responses, including vasodilation, plasma extravasation secondary to capillary leakage, edema, and mast cell degranulation. Convincing evidence obtained in rodent models suggests that neurogenic inflammation is assumed to contribute to the development of a migraine attack. Chemical stimulation of the dura mater triggers activation and sensitization of the trigeminal system and causes numerous molecular and behavioral changes; therefore, this is a relevant animal model of acute migraine. This narrative review discusses the emerging evidence supporting the involvement of neurogenic inflammation and neuropeptides in the pathophysiology of migraine, presenting the most recent advances in preclinical research and the novel therapeutic approaches to the disease.

## 1. Introduction

Migraine is a common neurological condition as the third most prevalent disease worldwide [1]. According to the Global Burden of Disease Study 2016, migraine is the second leading cause of disability [2]. The prevalence of migraine is 14.7%, and it is three times more common in women than men; in addition, women are less responsive to treatment [3]. Moreover, as migraine is a chronic episodic disorder that predominantly affects the working sector of a population, it thus has high social costs [4]. Migraine is ascribed to complicated, multifactorial conditions that give rise to substantial variations among patients and single patient responses to treatments [5]. Clinically, migraine can cause a variety of symptoms besides recurrent headaches, such as allodynia, photo- and phonophobia, and decreased daily activity, which can last from 4 to 72 h without treatment [6]. Due to its own pathogenesis and the fact that no other cause can be associated with the development of the disease, migraine belongs to the family of primary headache disorders [7].

The clinical course of migraine can be divided into different stages: the prodrome phase, a possible aura, followed by the headache, and the recovery stage (postdrome). The prodrome phase typically occurs up to a few days before the headache attack, and changes in well-being and behavior are experienced, while fatigue and impaired concentration occur as frequent complaints [8]. In 25% of the migraineurs, a temporary dysfunction of the central nervous system (CNS), the aura phenomenon, occurs [9]. The most common symptoms are visual (e.g., visual field disturbances), but sensory or speech disturbances and rarely motor symptoms can also be observed [6,10,11,12]. The typical aura appears before or at the beginning of the headache and lasts up to one hour. The headache in migraineurs is moderate or strong and throbbing, lasting 4–72 h, and is associated with sensitivity to light/sound and nausea/vomiting. Physical activity worsens the symptoms, and thus, the migraineurs seek rest (Figure 1A).

Despite intensive research, the pathomechanism of migraine is still unclear; however, activation and sensitization of the trigeminal system (TS) is essential during the attacks [13]. The TS is responsible for processing painful stimuli from the cortical area; during its activation, neurotransmitters, such as calcitonin gene-related peptide (CGRP), substance P (SP), pituitary adenylate cyclase-activating polypeptide (PACAP), and neurokinin A (NKA), are released both at the peripheral and central arm of the primary sensory neurons [14]. The neuropeptide release can induce mast cell degranulation and plasma extravasation, leading to neurogenic inflammation (NI) [15]. In the meantime, activation of the second-order neurons occurs in the caudal trigeminal nucleus (TNC) and their axons ascend to terminate in the thalamus, and the nociceptive information is projected to the primary somatosensory cortex [16]. Recent neuroimaging studies revealed other regions of the CNS (e.g., cerebellum, insula, pulvinar) that might play a role in the modulation of pain sensation [17,18] (Figure 1B).

In the 1950s, Ray and Wolff developed the first theory about the pathomechanism of migraine. They believed that migraine pain was caused by extracranial vasodilation, while intracranial vasospasm was responsible for aura symptoms [19]. At that time, this theory was in line with the pharmacological observations that the potent vasodilator amyl nitrate aborted the aura phase; meanwhile, ergotamine with vasoconstrictive properties decreased the headache [20]. Since vascular changes do not explain all the symptoms experienced during migraine attacks, new theories emerged regarding the pathomechanism of migraine.

The most widely accepted theory is focusing on the so-called cortical spreading depression (CSD) first described by Leao and Morison [21], which may be the equivalent of the aura phenomenon [22] playing a role in the development of migraine attacks [23]. During CSD, depolarization following an excitatory wave across the cerebral cortex changes cerebral blood supply, increases tissue metabolism, and releases amino acids and nitric oxide in the cortex, which activates nerves running in the dura, thus dilating the dural vessels, leading to sterile inflammation [24]. Under experimental conditions, CSD can activate secondary trigeminal nociceptors [25], suggesting that susceptibility to CSD might be responsible for the appearance of the attack.

Weiller and colleagues observed that the dorsolateral pons and the dorsal midbrain involving the nuclei nucleus raphe magnus (NRM), nucleus raphe dorsalis (DR), locus coeruleus (LC), and the periaqueductal grey matter (PAG) are activated during a migraine attack, which persists even after triptan treatment [26]. These nuclei can influence TNC activity, and they are involved in the transmission of pain. Brainstem serotonergic (NRM and PAG) and adrenergic (LC) nuclei contribute to the activation of the trigeminovascular system [27]. These brainstem areas, in addition to the trigeminovascular system, have a bidirectional connection with thalamus and hypothalamus. The thalamus has a role in integrating nociceptive inputs in migraine and pain sensation. The hypothalamus has direct connections with many structures involved in pain processing, including the nucleus tractus solitarius, rostral ventromedial medulla, PAG, and NRM [28,29,30]. It is hypothesized that the altered function of these brainstem migraine generators also plays a major role in attack development.

Nowadays, the most accepted concept is that migraine is a neurovascular disorder, which originates in the CNS, causing hypersensitivity to the peripheral trigeminal nerve fibers that innervate the vessels of the meninges. 

This narrative review discusses the processes underlying the pathomechanism of migraine, focusing on the role of neuropeptides and neurogenic inflammation. Furthermore, it emphasizes the importance of preclinical translational research, which has led to current understanding of migraine and summarizes the novel potential therapeutic options for migraine.

## 2. Dura Mater in Migraine

The dura mater, its vasculature supply, and the cerebral blood vessels are the only structures containing nociceptive nerve fibers [31]. The role of dura mater in migraine pain was widely examined. Ray and Wolff found that electrical stimulation of dural and cerebral vessels can cause nausea and the perception of headache-like pain in humans [19]. The dura mater is the outermost layer of the meninges and is located directly underneath the skull and vertebral column bones. The three branches of the trigeminal nerve (ophthalmic (V1), maxillary (V2), and mandibular (V3)) innervate the face and head region [32]. Dowgjallo and Grzybowski were the first to find the origin of meningeal nerve fibers in the trigeminal ganglion (TG) [33,34]. The tentorial nerve (a branch of the ophthalmic nerve) innervates most of the supratentorial dura; this nerve supplies the falx cerebri, calvarial dura, and superior surface of the tentorium cerebelli, forming a dense plexus with the arteries that form the vascular intracranial pain-sensitive structures [35]. The afferents innervating intracranial structures are collectively referred to as the trigeminovascular system [36,37]. Strassmann described that peripheral trigeminovascular neurons become mechanically hypersensitive to dural stimulation, which explains the pulsation and intensification of headache in case of cough or bending [38]. Furthermore, Burstein et al. observed that stimulation of the dura causes prolonged sensitization of central trigeminovascular neurons in the spinal trigeminal nucleus [39]. Several studies have shown peptidergic trigeminal afferents to innervate the dura mater [40,41,42].

## 3. Neuropeptides and Neurotransmitters

Meningeal nerve fibers are immunoreactive for CGRP, SP, NKA, neuropeptide Y (NPY), and vasoactive intestinal peptide (VIP), among others [43]. CGRP plays multiple roles in neurogenic inflammation [44]. In pharmacological and immunological experiments, antagonism of CGRP supported that CGRP is indirectly involved in plasma extravasation, which is primarily caused by SP and NKA [45]. Together with SP, CGRP can trigger mast cell degranulation to release proinflammatory and inflammatory compounds [46]. Beside these, dural mast cells and satellite glia express the CGRP receptor [47]. It is suggested that satellite glia and neurons are involved in a positive feedback loop of CGRP synthesis and release, maintaining increased inflammation and sensitization [48]. 

SP is widely distributed in the central and peripheral nervous systems of vertebrates [49]. In the CNS, it is present in the dorsal root ganglion, spinal cord, hippocampus, cortex, basal ganglia, hypothalamus, amygdala, and TNC [50,51] and has a role in the neurotransmission of pain and noxious stimuli in the spinal cord [52]. It has been described in numerous cell-type SP products, e.g., macrophages, eosinophils, lymphocytes, and dendritic cells [53,54]. The SP-induced release of inflammatory mediators, such as cytokines, oxygen radicals, and histamine, enhances tissue damage and stimulates further recruitment of leukocytes, thereby enhancing the inflammatory response [55]. SP induces local vasodilation and changes the vascular permeability, thereby increasing the delivery and accumulation of leukocytes into tissues to express local immune responses [56]. SP often co-expresses with other transmitter molecules, like CGRP and glutamate in the TG and trigeminal nucleus caudalis [57,58]. During the headache phase of migraine, a significant increase in plasma SP and CGRP levels is demonstrated [59]. 

Moreover, PACAP is found in several structures that are relevant to the pathomechanism of migraine, e.g., in the dura mater, the cerebral vessels [60], the TG [61], the TNC [62], and the cervical spinal cord [63]. It was recently found that PACAP is co-expressed with CGRP in some dural nerve fibers [64]. PACAP plays a role in neuromodulation, neurogenic inflammation, and nociception [65], and in addition, it is involved in the higher-order processing of pain in brain regions such as the thalamus and the amygdala [66,67]. PACAP is also relevant in the central sensitization and emotional load of pain [68]. Zhang and colleagues found that following inflammation in sensory neurons, PACAP is upregulated [69]. Meningeal sensory fibers can release neuropeptides from their peripheral endings in the meninges, where they can evoke components of neurogenic inflammation [64].

VIP is widely distributed in the central and peripheral nervous systems [70]. VIP plays as potent vasodilators, acting on the smooth muscle cells in arterioles [71]. VIP can modulate mast cell degranulation and the production of proinflammatory cytokines, such as interleukins, including IL-6 and IL-8 [72]. In a clinical study, during the interictal period of chronic migraine, higher VIP levels have been reported in peripheral venous blood than in control subjects [73]. Pellesi et al. observed that as opposed to shorter vasodilation, prolonged VIP-mediated vasodilation causes more headaches [74]. Together, VIP may contribute to migraine pain through vasodilation and dural mast cell degranulation.

Transient receptor potential vanilloid-1 receptor (TRPV1), a nonselective cation channel, is a molecular component of pain detection and modulation [75]. TRPV1 receptors are present in the human TG [76] and trigeminal afferents, which innervate the dura mater [77]. In addition to excessive heat, various exogenous and endogenous triggering factors can directly activate or sensitize TRPV1 [78]. TRPV1 activation leads to the release of neuropeptides, such as SP and CGRP, which can cause vasodilation and initiate neurogenic inflammation within the meninges [79]. TRPV1 activation and/or sensitization can enhance inflammatory responses via the expression and release of other inflammatory mediators.

Histamine plays a role in migraine; it can modulate neurogenic inflammation and nociceptive sensitization [80]. During a migraine attack, elevated levels of a histamine precursor histidine were found in plasma and cerebrospinal fluid (CSF) [81], and the histamine levels of the plasma were increased both ictally and interictally in migraine patients [80]. The release of SP contributes to local vasodilation, induces histamine release from mast cells, and produces flare and further activates other sensory nerve endings [82]. C-fibers are known to be activated by histamine and are responsible for the neuropeptide release. Nerve fibers, which contain histamine, have been found in the superficial laminae of the dorsal horn, an essential site for nociceptive transmission [83]. In inflammatory conditions, histamine can mediate the release of SP and glutamate [84] (Table 1).

## 4. Neurogenic Inflammation

The localized form of inflammation is neuroinflammation, which occurs in both the peripheral and CNSs. The main features of NI are the increased vascular permeability, leukocyte infiltration, glial cell activation, and increased production of inflammatory mediators, such as cytokines and chemokines [85]. NI increases the permeability of the blood to the brain barrier, thus allowing an increased influx of peripheral immune cells into the CNS [86] (Figure 2A).

The concept of NI was introduced by the experiment of Goltz and Bayliss, in which skin vasodilation was observed during electrical stimulation of the dorsal horn, which could not be linked to the immune system [87,88]. Dalessio was the first who hypothesized a connection between NI and migraine and believed that a headache is a result of vasodilation of cranial vessels associated with local inflammation [89]. This theory was later reworked by Moskowitz, who believed that upon activation, the neuropeptide release from trigeminal neurons has a role in increasing vascular permeability and vasodilation [90].

There are several theories concerning the mechanism of NI. Hormonal fluctuations or cortical spreading depression can initiate two types of processes: activating the TS to trigger the liberation of neuropeptides from the peripheral trigeminal afferents and/or degranulating the mast cells that can lead to the release of neuropeptides by activating and sensitizing the nociceptors [91]. In rats, Bolay and colleagues demonstrated that after local electrical stimulation of the cerebral cortex, CSD is generated, and it can trigger trigeminal activation, which causes meningeal inflammation occurring after the CSD disappearance [92]. Both CGRP and SP play an important role in the development of NI. Released peptides, such as CGRP, bind to its receptor on smooth muscle cells, eliciting a vasodilatory response, thereby increasing meningeal blood flow in the dural vasculature. In contrast to CGRP, binding of the released SP to the NK1 receptors expressed on the microvascular blood vessels disrupts the membrane and causes plasma protein leakage. Both neuropeptides can induce mast cell degranulation through their specific receptors and further sensitize meningeal nociceptors [91]. The meningeal nerve fibers also contain neurotransmitters (e.g., glutamate, serotonin) and hormones (e.g., prostaglandins) that can affect the activation and release of neuropeptides, causing neurogenic inflammation [89]. Moreover, several cell types (e.g., endothelial cells, mast cells, and dendritic cells) can release tumor necrosis factor alpha (TNFα), interleukins, nerve-growth factor (NGF), and VIP, also causing plasma protein extravasation (PPE) [93,94], which is a key characteristic of NI. In addition, neuronal nitric oxide synthase (nNOS) enzyme can be detected in the trigeminal nerve endings, the dural mastocytes, and also the TNC and the TG [95], which catalyzes the synthesis of retrograde signaling molecule nitric oxide (NO). NO has a major role in mediating many aspects of inflammatory responses; NO can affect the release of various inflammatory mediators from cells participating in inflammatory responses (e.g., leukocytes, macrophages, mast cells, endothelial cells, and platelets) [96]. Through its retrograde signaling action, astrocytes can influence the release of CGRP, SP, and glutamate [97,98]. Beside this, bradykinin and histamine induce NO release from vascular endothelial cells, suggesting a strong interaction between NO and inflammation [99]. The inflammation can lead to CGRP release from the activated primary afferent neurons, which force satellite glial cells to release NO. NO can induce nNOS, which can be considered a significant marker of the sensitization process of the TS. TRPV channels permit afferent nerves to detect thermal, mechanical, and chemical stimuli, thereby regulating NI and nociception [100]. TRPV1 was identified in dorsal root ganglion (DRG), TG neurons, and spinal and peripheral nerve terminals [101]. Inflammatory mediators remarkably up-regulate TRPV1 through activation of phospholipase C (PLC) and protein kinase A (PKA) and protein kinase C (PKC) signaling pathways [102,103,104,105]. Increased TRPV1 expression in peripheral nociceptors contributes to maintaining inflammatory hyperalgesia [101]. In an experimental injury model, Vergnolle et al. demonstrated that a decrease in osmolarity of extracellular fluid could induce neurogenic inflammation, which TRPV4 can mediate [106]. Furthermore, plasma and cerebrospinal fluid levels of neuropeptides, histamine, proteases, and pro-inflammatory cytokines (e.g., TNFα, IL-1β) are elevated during migraine attacks [107,108], suggesting that neuroimmune interactions contribute to migraine pathogenesis.

### 4.1. Vasodilation

There are various cell types in blood vessels that both release and respond to numerous mediators that can contribute to migraine; this includes growth factors, cytokines, adenosine triphosphate (ATP), and NO [109,110,111,112]. In the central system, NO may be involved in the regulation of cerebral blood flow and neurotransmission [59]. NO can stimulate the release of neuropeptides and causes neurogenic vasodilation [113]. In addition to NO, NGF also increases the expression of CGRP and enhances the production and release of neuropeptides, including SP and CGRP, in sensory neurons [114]. CGRP, a potent vasodilator, is released from intracranial afferents during migraine attacks. This vasodilatory effect of CGRP is mediated by its action on CGRP receptors, which stimulates the adenyl cyclase and increases cyclic adenosine monophosphate (cAMP), thus producing potent vasodilation via the direct relaxation of vascular smooth muscle [115,116]. In response to prolonged noxious stimuli, SP is released from trigeminal sensory nerve fibers around dural blood vessels, leading to endothelium-dependent vasodilation [82]. VIP also contributes to neurogenic inflammation by inducing vasodilation [117] (Figure 2B).

### 4.2. Plasma Protein Extravasation

Another critical feature of neurogenic inflammation is PPE. Based on preclinical studies, the neurogenic PPE plays a role in the pathogenesis of migraine [118]. In several studies, following electrical stimulation of the trigeminal neurons or intravenous capsaicin, the peripheral nerve endings in the dural vasculature released SP, which caused plasma protein leakage and vasodilation through the NK-1 receptors [119]. Transduction of the SP signal through the NK1 receptor occurs via G protein signaling and the secondary messenger cAMP, ultimately leading to the regulation of ion channels, enzyme activity, and alterations in gene expression [120]. SP can indirectly influence plasma extravasation by activating mast cell degranulation, which results in histamine release [83]. In addition, NKA is able to induce plasma protein efflux and activate inflammatory cells [121] (Figure 2C).

### 4.3. Mast Cell Degranulation

It is well known that dural mast cells play a role in the pathophysiology of migraine [122]. Meningeal mast cells are in close association with neurons, especially in the dura, where they can be activated following trigeminal nerve and cervical or sphenopalatine ganglion stimulation [123]. The release of neuropeptides, such as CGRP, PACAP, and SP, from meningeal nociceptors can cause the degranulation of mast cells [124], resulting in the release of histamine and serotonin and selectively can cause the release of pro-inflammatory cytokines, such as TNF-α, IL-1, and IL-6 [125,126,127]. The plasma and CSF levels of these mediators (e.g., CGRP, TNFα, and IL-1β) are enhanced during migraine attacks [104]. VIP promotes degranulation of mast cells [128], similar to the effects of SP [83]. It was found that CSD can induce intracranial mast cell degranulation and promote the activation of meningeal nociceptors [129,130]. Besides these, according to several studies, mast cells can be activated by acute stress [123,131,132], which is known to precipitate or exacerbate migraines [133,134]. Based on these findings, mast cells in themselves may promote a cascade of associated inflammatory events resulting in trigeminovascular activation (Figure 2D).

### 4.4. Microglia Activation

Microglia appears in the CNS and can exert neuroprotective and neurotoxic effects as well. Under the influence of inflammatory stimuli, microglia can become efficient mobile effector cells [135]. Microglia activation leads the production of inflammatory mediators and cytotoxic mediators (e.g., NO, reactive oxygen species, prostaglandins) [136,137], which might disrupt the integrity of the blood brain barrier, thereby allowing leukocyte migration into the brain [138]. Microglia express receptors for neurotransmitters, such as glutamate, gamma- aminobutyric acid, noradrenaline, purines, and dopamine [139]. It has been described that activation of ion channels is related to the activation of microglia; therefore, neurotransmitters probably influence microglia function [140]. Glutamate leads to neuronal death but is also an activation signal for microglia [141]. Activation of glutamate receptors causes the release of TNF-α, which, with microglia-derived Fas ligand, leads to neurotoxicity [142]. Besides this, Off signals from neurons appear important in maintaining tissue homeostasis and limiting microglia activity under inflammatory conditions, presumably preventing damage to intact parts of the brain [143]. Endothelin B-receptor-mediated regulation of astrocytic activation was reported to improve brain disorders, such as neuropathic pain [144]. SP also directly activates microglia and astrocytes and contributes to microglial activation [145,146], initiating signaling via the nuclear factor kappa B pathway, leading to pro-inflammatory cytokines production [147] (Figure 2E).

### 4.5. Cytokines, Chemokines

Cytokines are small proteins produced by most cells in the body, which possess multiple biologic activities to promote cell-cell interaction [148]. There is evidence that cytokines play an important role in several physiological and pathological settings, such as immunology, inflammation, and pain. [149]. The most important pro-inflammatory cytokines include IL-1, IL-6, and TNFα, and the key chemokine is IL-8 [149]. Cytokines and chemokines are released by neurons, microglia, astrocytes, macrophages, and T cells, and these factors might activate nociceptive neurons [150]. TNFα can trigger tissue edema and immune cell infiltration [151] and can influence the reactivity of signal nociceptors to the brain and increase blood levels during headaches, playing a crucial role in the genesis of migraine [152]. Cytokines are considered to be pain mediators in neurovascular inflammation, which generates migraine pain [153]. They can induce sterile inflammation of meningeal blood vessels in migraines [154]. Besides this, elevated levels of chemokines can stimulate the activation of trigeminal nerves and the release of vasoactive peptides; thereby, they can induce inflammation [155]. Based on these, cytokines and chemokines might contribute to migraine.

## 5. Animal Models of Neurogenic Inflammation

Developing animal models of human illnesses is a challenging task for translational research, but it is indispensable to understanding pathomechanism, searching for biomarkers, and engineering novel treatment [156,157,158,159,160,161,162,163,164]. Migraine research is no exception. Chemical activation of meningeal trigeminal nociceptors is possible in animal experiments. The use of Complete Freund’s adjuvant (CFA, dried and inactivated Mycobacterium tuberculosis in mineral oil) or inflammatory soup (IS, a standard mixture of histamine, serotonin, bradykinin, and prostaglandin E2) on the surface of the dura mater is a useful method for inducing trigeminal activation and sensitization and developing neurogenic inflammation in rats [165,166,167]. It has been shown that trigeminal brainstem neurons have been sensitive to both subarachnoid superfusion and topical IS administration [37,168]. Lukács et al. demonstrated that the application of CFA or IS onto the dural surface can induce changes in the expression of phosphorylated extracellular signal-regulated kinase ½ (pERK1/2), IL-1β, and CGRP-positive nerve fibers in the TG [167]. Similar to the previous experiment, Laborc et al. used topical administration of IS or CFA on the dura mater to examine the activation pattern that is caused by chemicalstimulation, and they found that application of IS on the dura mater induces short-term c-Fos activation, while CFA did not cause any difference in the number of c-Fos-positive cells between the CFA-treated and control groups. Whereas short survival times were used, the authors believe this may have been the reason that the CFA did not prove effective [169]. Spekker et al. found that IS was able to cause sterile neurogenic inflammation in the dura mater and increased the area covered by CGRP and TRPV1 immunoreactive fibers and the number of neuronal nitric oxide synthase (nNOS)-positive cells in the TNC, and pretreatment with sumatriptan or kynurenic acid (KYNA) could modulate the changes caused by IS. Sumatriptan probably acted through the 5-HT_1B/1D_ receptors, while KYNA possibly acted predominantly by inhibiting the glutamate system and thereby blocking sensitization processes, which is important in migraine [170]. Furthermore, Wieseler and colleagues observed an increase in the level of IL-1β and TNFα, and the microglial/macrophage activation marker CD11b in Sp5C after IS was administered bilaterally through supradural catheters in freely moving rats [171].

In addition to morphological changes, IS can also influence animal behavior. Oshinsky and Gomonchareonsiri used IS treatment three times per week for up to four weeks, and they demonstrated that repeated infusions of IS over weeks induced a long-lasting decrease in periorbital pressure thresholds [172]. Melo-Carrillo and colleagues described that repeated infusion of IS increased the resting and freezing behavior and decreased the locomotor activity [173]. These observations are consistent with decreased routine physical activity and lack of exercise due to migraine-induced pain in migraine patients [174]. Moreover, they found a specific ipsilateral facial grooming behavior, which may be related to the unilateral nature of migraine. In an animal model of intracranial pain, Malick et al. showed that simultaneous chemical and mechanical stimulation of the dura mater not only increases the number of Fos-positive neurons in the medullary dorsal horn but can reduce the appetite of the rats. [175]. Wieseler et al. experienced facial and hind paw allodynia and after two IS infusions [176]. In a novel large animal model of recurrent migraine, repeated chemical stimulation of the dura mater reduced locomotor behavior, which may mimic a decrease in routine physical activity in people with headaches. In addition, increased scratches and slow movements were observed; these may reflect pain localized to the head area [177] (Figure 3). Based on these experiments, it can be said that dural application of IS triggers activation and sensitization of the trigeminal system. Therefore, this is a relevant animal model of acute migraine.

## 6. Current Treatments and Advances in Preclinical Research

Triptans are widely used to relieve migraine attacks; acting as agonists on 5-hydroxytryptamine receptors (5-HT_1B/1D_), they can cause the constriction of dilated cranial arteries and the inhibition of CGRP release [178]. In an animal model of migraine, after electrical stimulation of the TG, sumatriptan attenuates PPE by preventing the release of CGRP [179]. In knockout mice and guinea pigs, it has been shown that 5-HT_1D_ receptors have a role in the inhibition of neuropeptide release, thereby modifying the dural neurogenic inflammatory response [180]. The use of triptans is limited by their vasoconstrictive properties. As triptans are not effective in everyone, they often lead to medication overuse, triggering migraine to become chronic (Table 2). 

Ditans target the 5-HT_1F_ receptor, which is expressed in the cortex, the hypothalamus, the trigeminal ganglia, the locus coeruleus, the middle cerebral artery, and the upper cervical cord. Lasmiditan is the first drug approved for clinical use. Contrary to triptans, Lasmiditan does not cause vasoconstriction. The activation of 5-HT_1F_ receptor inhibits the release of CGRP and probably SP from the peripheral trigeminal endings of the dura and acts on the trigeminal nucleus caudalis or the thalamus [181].

Besides triptans and ditans, acute treatments of migraine headaches, i.e., ergot alkaloids and nonsteroidal anti-inflammatory agents (NSAIDs), may decrease the neurogenic inflammatory response [182]. NSAIDs have anti-inflammatory, analgesic, and anti-pyretic properties. Their primary effect is to block the enzyme cyclooxygenase and hence mitigate prostaglandin synthesis from arachidonic acid [183]. Both acetaminophen and ibuprofen, which can reduce pain intensity, can also be used in children. Magnesium pretreatment increases the effectiveness of these treatments and reduces the frequency of pain [184]. Ergotamine has been recommended to abort migraine attacks by eliminating the constriction of dilated cranial blood vessels and carotid arteriovenous anastomoses, reducing CGRP release from perivascular trigeminal nerve endings, and inhibit the nociceptive transmission on peripheral and central ends of trigeminal sensory nerves [185]. 

An alternative treatment strategy is the use of CGRP-blocking monoclonal antibodies. Monoclonal antibodies have a number of positive properties: (1) a long half-life, (2) long duration of action, and (3) high specificity [186]. Four monoclonal antibodies are currently developing for migraine prevention: three against CGRP and one against the CGRP receptor. The safety and tolerability of these antibodies are promising; no clinically significant change in vitals, ECGs, or hepatic enzymes was observed. Blocking of CGRP function by monoclonal antibodies has demonstrated efficacy in the prevention of migraine with minimal side effects in multiple Phase II and III clinical trials [187]. 

Another alternative approach to treating the migraine attack by limiting neurogenic inflammatory vasodilation is the blockade of CGRP receptors by selective antagonists. Gepants were designed to treat acute migraines [188]. These bind to CGRP receptors and reverse CGRP-induced vasodilation but were not vasoconstrictors per se [189]. Based on these, gepants may be an alternative if triptans are contraindicated. Currently, two gepants (Ubrogepant, Rimegepant) are available on the market, but several are in development. 

In gene-knockout studies, the hypothesis the tachykinins are the primary mediators of the PPE component of NI has been strengthened [190,191]. Following topical application of capsaicin to the ear, the PPE was decreased in Tac1-deficient mice compared to wild-type mice [192]. Following activation of the trigeminal system by chemical, mechanical, or electrical stimulation, tachykinin Receptor 1 (TACR1) antagonists seem to be adequate to blocking dural PPE [193]. However, lanepitant, a selective TACR1 antagonist, has no significant effect on migraine-associated symptoms [194]; moreover, it was found ineffective in a migraine prevention study [195]. The only currently available and clinically approved NK1 receptor antagonist is aprepitant, which is used as an antiemetic to chemotherapy-induced nausea in cancer patients [196]. 

In animal models, blockage of TRPV1 receptors was effective to reverse inflammatory pain; however, TRPV1 antagonists produce some serious side effects, e.g., hyperthermia [197]. Clinical data suggest that TRPV1 antagonists might be effective as therapeutic options for certain conditions, such as migraine and pain related to several types of diseases. Hopefully, current clinical trials with TRPV1 receptor antagonists and future studies provide an answer as to the role of TRPV1 in inflammatory and neuropathic pain syndromes. 

The anti-nociceptive effects of endocannabinoids are thought to be mediated mainly through the activation of cannabinoid receptor type 1 (CB1) [198]. Localization of CB1 receptors along the trigeminal tract and trigeminal afferents [199,200] suggests that the endocannabinoid system can modulate the neurogenic-induced migraine [201]. Clinical data suggested that in migraine patients, the endocannabinoid levels are lower [202,203]. In animal models of migraine, endocannabinoids can reduce neurogenic inflammation. Akerman et al. reported that capsaicin-induced vasodilation is less after intravenous administration of anandamide (AEA); in addition, AEA significantly prevented CGRP- and NO-induced vasodilation in the dura [204]. In a previous study, Nagy-Grócz and colleagues observed that NTG and AEA alone or combined treatment of them affects 5-HTT expression, which points out a possible interaction between the serotonergic and endocannabinoid system on the NTG-induced trigeminal activation and sensitization phenomenon, which are essential during migraine attacks [205]. These results raise the possibility that the AEA has a CB1 receptor-mediated inhibitory effect on neurogenic vasodilation of trigeminal blood vessels. Based on these, anandamide may be a potential therapeutic target for migraine. Besides these, the presence of CB1 receptors in the brain makes them a target for the treatment of migraine, blocking not only peripheral but also central nociceptive traffic and reducing CSD. CB2 receptors in immune cells may be targeted to reduce the inflammatory component associated with migraine.

PACAP and its G-protein-coupled receptors, pituitary adenylate cyclase 1 receptor (PAC1) and vasoactive intestinal peptide receptor 1/2 (VPAC1/2), are involved in various biological processes. Activation of PACAP receptors has an essential role in the pathophysiology of primary headache disorders, and PACAP plays an excitatory role in migraine [206]. There are two pharmacology options to inhibit PACAP: PAC1 receptor antagonists/antibodies directed against the receptor or antibodies directed against the peptide PACAP [207]. Studies of the PAC1 receptor antagonist PACAP (6–38) have proved that antagonism of this receptor may be beneficial even during migraine progression [208]. PACAP38 and PAC1 receptor blockade are promising antimigraine therapies, but results from clinical trials are needed to confirm their efficacy and side effect profile. 

The tryptophan-kynurenine metabolic pathway (KP) is gaining growing attention in search of potential biomarkers and possible therapeutic targets in various illnesses, including migraine [209,210]. KYNA is a neuroactive metabolite of the KP, which affects several glutamate receptors, playing a relevant role in pain processing and neuroinflammation [181]. KYNA may block the activation of trigeminal neurons, affect the migraine generators, and modulate the generation of CSD [209,211]. An abnormal decrease or increase in the KYNA level can cause an imbalance in the neurotransmitter systems, and it is associated with several neurodegenerative and neuropsychiatric disorders [212,213,214,215]. Based on human and animal data, the KP is downregulated under different headaches; thus, possibly less KYNA is produced [216]. It is consistent with the theory of hyperactive NMDA receptors, which play a key role in the development of central sensitization [217] and thus in migraine pathophysiology. In an NTG-induced rodent model of migraine, Nagy-Grócz et al. demonstrated a decrease in the expression of KP enzymes after NTG administration in rat TNC [218]. Interferons can control the transcription expression of indoleamine 2,3-dioxygenase (IDO), kynurenine 3-monooxygenase (KMO), and kynureninase (KYNU); therefore, the pro-inflammatory cytokines may affect the kynurenine pathway [219]. It is difficult for KYNA to cross the blood-brain barrier (BBB); therefore, synthetic KYNA analogs may provide an additional alternative for synthesizing compounds that have neuroprotective effects comparable to KYNA that can cross the BBB effectively. Preclinical studies have shown the effectiveness of KYNA analogs in animal models of dural stimulation [220,221]. Further preclinical studies are required to understand the role of KYNA analogs in migraine and clinical studies that assess their effectiveness in acute or prophylactic treatment (Figure 4).

In animal models of chronic pain and inflammation and several clinical trials, palmitoylethanolamide (PEA), endogenous fatty acid amide, has been influential on various pain states [222,223,224]. In a pilot study, for patients suffering from migraine with aura, ultra-micronized PEA treatment has been shown effective and safe [225]. Based on these, PEA is a new therapeutic option in the treatment of pain and inflammatory conditions.

## 7. Conclusions and Future Perspective

Migraines impose a tremendous negative impact on quality of life; nevertheless, antimigraine pharmacotherapy provides limited success in efficacy and tolerability. Migraineurs and patients with chronic pain helplessly seek alternative or complementary treatments, such as biofeedback, botox, yoga, acupuncture, acupressure, and music therapy, among others [226]. The biggest challenge in antimigraine research may lie in complex multifactorial pathogenesis of migraine headache, which is precipitated by interwinding genetic, endocrine, metabolic, and/or environmental factors, and thus, the exact pathology leading to migraine attack remains poorly understood. This review article focuses on that migraine headache is a reflection of neurogenic inflammation in the activation and sensitization of trigeminovascular afferent nerves, which project to the second-order neurons in the brainstem. The local release of neuropeptides and neurotransmitters can not only cause the dilation of meningeal vessels but also induce neuroinflammation. Animal models of migraine support this hypothesis that neurogenic inflammation plays a crucial role in the sensitization process that leads to enhanced responsiveness of target tissues. However, clinical study remains to be conducted. Understanding the signal transduction and regulation of neuropeptides, including CGRP, SP, or neurokinin A, may open an approach to discovery of new targets leading to the prevention of neurogenic inflammation.

Moreover, NI can be initiated by chronic stress, diet, hormonal fluctuations, or CSD. The NI-triggering factors may become a possible interventional target preventing the initiation of neuroinflammatory cascade. Immune reactions can also participate in NI. However, little is known about the interaction of the immune system in NI. Understanding the mechanism of NI trigger is essential in migraine research. Migraine headache is frequently observed in patients with cardiovascular diseases, respiratory diseases, psychiatric diseases, and restless legs syndrome. The disturbance of the serotonergic nervous system, the sympathetic nervous system, and the hypothalamic-pituitary-adrenal axis links migraine to mood disorders and obesity. Thus, identifying predisposing factors precipitating to the NI trigger may be a potential clue for a novel approach of migraine treatment.

Identification and usage of specific disease biomarkers can be suitable to guide the treatment and monitor the improvement or worsening of migraine symptoms during the treatment. MicroRNAs (miRs) may be useful as biomarkers of several diseases, including pain conditions and migraine in both adults and children. Deregulation of miRNAs has recently been described in migraine patients during attacks and pain-free periods [227]. In addition, significant levels of some miRs have been demonstrated in the serum of migraine children and adolescents without aura [228], suggesting that they are involved in the pathogenetic mechanisms of migraine, further enhancing the role of these miRs in the pathophysiology of migraine and their potential use as potential biomarkers.

We are untangling the puzzle of the mechanisms behind migraine attacks. However, finding the initial cause and effective treatment remains far away. The translational animal research currently tows forward the field of migraine research and may successfully serve as a savior of migraineurs in the future.

## Figures and Tables

**Figure 1 biomedicines-10-00076-f001:**
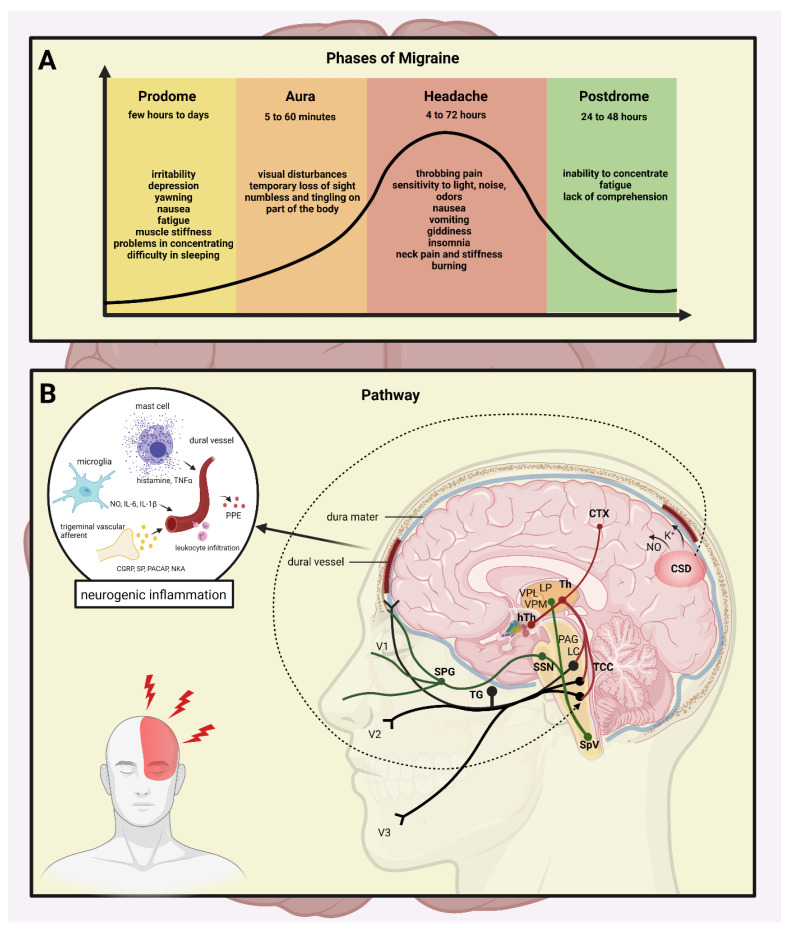
The stages and the pathways of migraine. (**A**) The stages of migraine attack: the prodrome phase, a possible aura, followed by the headache, and subsequently the postdrome. A strong headache is frequently accompanied with nausea, vomiting, and sensitivity to light, which lasts 4 to 72 h. (**B**) Mechanisms and structures involved in the pathogenesis of migraine: CTX, cortex; NO, nitric-oxide; CSD, cortical spreading depression; Th, thalamus; hTh, hypothalamus; LP, lateral posterior nucleus; VPM, ventral posteromedial nucleus; VPL, ventral posterolateral nucleus; PAG, periaqueductal grey matter; LC, locus coeruleus; TCC, trigeminocervical complex; SSN, superior salivatory nucleus; SpV, spinal trigeminal nucleus caudalis; TG, trigeminal ganglion; SPG, sphenopalatine ganglion; V1, ophthalmic nerve; V2, maxillary nerve; V3, mandibular nerve; CGRP, calcitonin gene-related peptide; SP, substance P; PACAP, pituitary adenylate cyclase-activating polypeptide; NKA, neurokinin A; PPE, plasma protein extravasation; TNFα, tumour necrosis factor alpha; IL, interleukin.

**Figure 2 biomedicines-10-00076-f002:**
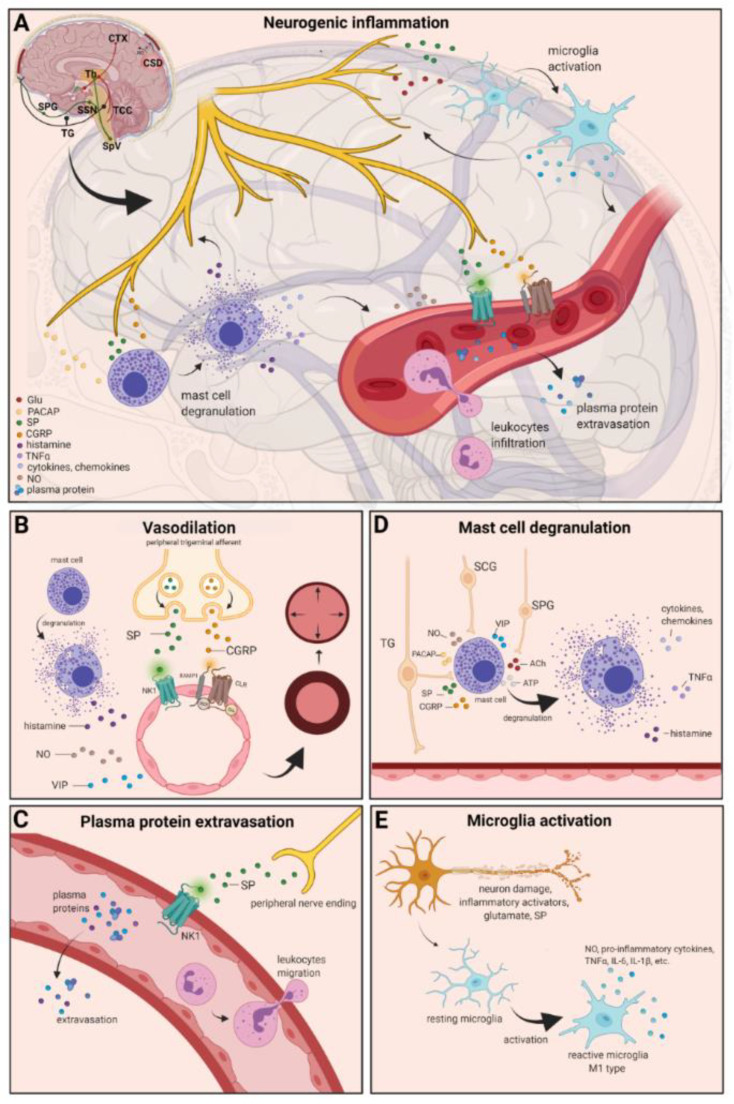
Neurogenic inflammation and its main features. (**A**) Stimulation of the trigeminal nerve causes the release of neuropeptides, including CGRP, SP, NO, VIP, and 5-HT, leading to neurogenic inflammation, which has four main features: the increased vascular permeability, leukocyte infiltration, glial cell activation, and increased production of inflammatory mediators, such as cytokines and chemokines. (**B**) Vasoactive peptides, such as CGRP and SP, bind their receptors on smooth muscle of dural vessels and cause vasodilation. The released neuropeptides induce mast cell degranulation, resulting in the release of histamine, which leads endothelium-dependent vasodilation. (**C**) Binding of the released SP to the NK1 receptors expressed on the microvascular blood vessels disrupts the membrane and causes plasma protein leakage and leukocyte extravasation. (**D**) Mast cells are in close association with neurons, especially in the dura, where they can be activated following trigeminal nerve and cervical or sphenopalatine ganglion stimulation. Release of neuropeptides causes mast cell degranulation, which leads to release of histamine and serotonin and selectively can cause the release of pro-inflammatory cytokines, such as TNF-α, IL-1, and IL-6. (**E**) Under the influence of inflammatory stimuli, microglia can become reactive microglia. Microglia activation leads to the production of inflammatory mediators and cytotoxic mediators.

**Figure 3 biomedicines-10-00076-f003:**
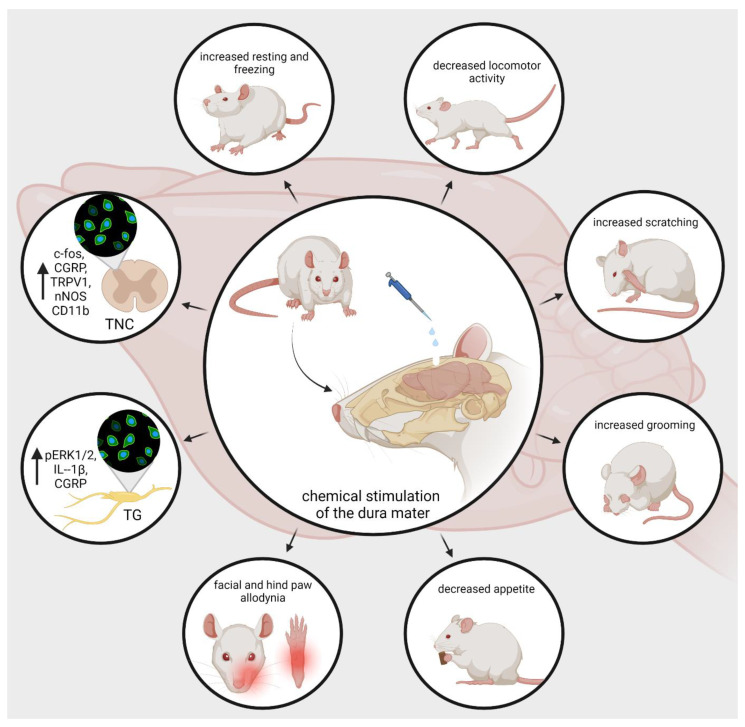
nNOS, pERK, ILs, increase the resting and freezing behavior, and decrease the appetite and locomotor activity of the animals. In addition, it can enhance grooming and scratching behavior and elicit mechanical and thermal hypersensitivity. CGRP, calcitonin gene-related peptide; TRPV1, transient receptor potential vanilloid receptor; nNOS, neuronal nitric-oxide synthase; IL, interleukins; pERK, phosphorylated extracellular signal-regulated kinase.

**Figure 4 biomedicines-10-00076-f004:**
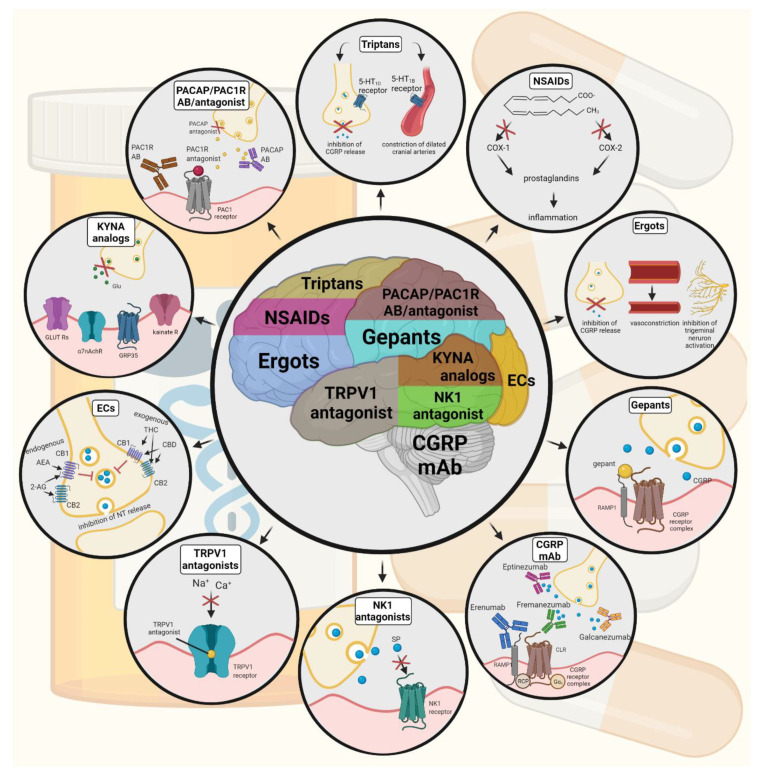
Possible treatments of neurogenic inflammation and migraine. NSAIDs, non-steroidal anti-inflammatory drugs; 5-HT, serotonin; CGRP, calcitonin gene-related peptide; COX, cyclooxygenase; Ab, antibody; NK1, neurokinin 1; TRPV1, transient receptor potential vanilloid receptor; SP, substance P; EC, endocannabinoids; AEA, anandamide; 2-AG, 2-arachidonoylglycerol; CB, cannabinoid receptor; THC, tetrahidrokanabinol; CBD, cannabidiol; NT, neurotransmitter; GLUT R, glutamate receptors; α7AchR, alpha-7 nicotinic receptor; GPR35, G protein-coupled receptor 35; PACAP, pituitary adenylate cyclase-activating polypeptide; PAC1R, pituitary adenylate cyclase 1 receptor.

**Table 1 biomedicines-10-00076-t001:** Neuropeptides and neurotransmitters and their role in migraine and neurogenic inflammation.

Neuropeptides/Neurotransmitters	Receptors	Migraine/Neurogenic Inflammation-Related Functions	References
CGRP	CLR, RAMP1	craniocervical vasodilation, peripheral and central sensitization, neuron-glia interaction, involved plasma extravasation, mast cell degranulation,	Asghar et al., 2011 [44], Holzer, 1998 [45], Ottosson and Edvinsson, 1997 [46], Lennerz et al., 2008 [47], Raddant and Russo, 2011 [48]
SP	NK1	craniocervical vasodilation, plasma protein extravasation, cytokines, oxygen radicals, and histamine release	Hökfelt et al., 1975 [49], Ribeiro-da-Silva and Hökfelt, 2000 [50], Snijdelaar et al., 2000 [51], Graefe and Mohiuddin, 2021 [52], Killingsworth et al., 1997 [53], Weinstock et al., 1988 [54], Holzer and Holzer-Petsche, 1997 [55], Pernow, 1983 [56], Gibbins et al., 1985 [57], Battaglia and Rustioni, 1988 [58], Malhotra, 2016 [59]
PACAP	PAC1, VPAC1, VPAC2	craniocervical vasodilation, peripheral and central sensitization	Jansen-Olesen and Hougaard Pedersen, 2018 [60], Eftekhari et al., 2013 [61], Nielsen et al., 1998 [62], Jansen-Olesen et al., 2014 [63], Uddman et al., 2002 [64], Hashimoto et al., 2006 [65], Martin et al., 2003 [66], Missig et al., 2014 [67], Kaiser and Russo, 2013 [68], Zhang et al., 1998 [69]
VIP	VPAC1, VPAC2	craniocervical vasodilation, mast cell degranulation, IL-6 and IL-8 production	Kilinc et al., 2015 [70], Ohhashi et al., 1983 [71], Kakurai et al., 2001 [72], Cernuda-Morollón et al., 2014 [73], Pellesi et al., 2020 [74]
-	TRPV1	vasodilation, peripheral and central sensitization, neuropeptide release (SP, CGRP) initiate neurogenic inflammation	Caterina et al., 2000 [75], Quartu et al., 2016 [76], Dux et al., 2020 [77], Bevan et al., 2014 [78], Meents et al., 2010 [79]
histamine	H_1–4_R	vasodilation, mediate SP and glutamate release	Yuan and Silberstein, 2018 [80], Castillo et al., 1995 [81], Heatley et al., 1982 [82], Foreman et al., 1983 [83], Rosa and Fantozzi, 2013 [84]

**Table 2 biomedicines-10-00076-t002:** Current treatments and advances in preclinical research.

Drug Class	Drug	Target	FDA Appoved
NSAIDs	Acetylsalicylic acid	COX1–2	yes
	Ibuprofen		yes
	Diclofenac potassium		yes
	Paracetamol		yes
Triptans	Sumatriptan	5-HT_1D_ receptor	yes
	Zolmitriptan		yes
	Almotriptan		yes
	Rizatriptan		yes
	Frovatriptan		yes
	Naratriptan		yes
	Eletriptan	5-HT_1B/1D_ receptor	yes
Ditans	Lasmiditan	5-HT_1F_ receptor	yes
Gepants	Ubrogepant	CGRP receptor	yes
	Rimegepant		yes
	Atogepant		no
	Vazegepant		no
Ergot alkaloids	Ergotamine tartrate	α-adrenergic receptor,5-HT receptors	yes
CGRP/CGRP receptor monoclonal antibody	Erenumab	CGRP receptor	yes
	Eptinezumab	CGRP ligand	yes
	Fremanezumab		yes
	Galcenezumab		yes
NK1R antagonists	Aprepitant	NK1 receptor	yes
PACAP/PAC1 receptor monoclonal antibody	ALD1910	PACAP38	no
	AMG-301	PAC1 receptor	no
Endocannabinoids	2-Arachidonoylglycerol	CB1 receptor	no
	Anandamide	CB1 receptor	no

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
