# Peer review of "Neurogenic Inflammation: The Participant in Migraine and Recent Advancements in Translational Research"

_biomedicines, 2021, doi:10.3390/biomedicines10010076_

Round 1
Reviewer 1 Report
Dear Authors,
I have read the manuscript and I send you my comments:
1) Methods are missing, please add the methods used to choose the manuscript
2) Please add a table for each section in order to do more easy the lecture
3) section 5 animal studies, this section can be deleted with the related figure
4) section 6: treatment page 12 regarding NSAIDs these drugs are commonly used in the management of headache also in chieldren so add more data regarding these drugs (see the manuscript "Headache. 2014 Feb;54(2):313-24. doi: 10.1111/head.12162." Please add also the role of PEA in the management of headache (see Front Neurol. 2018 Aug 17;9:674. doi: 10.3389/fneur.2018.00674. eCollection 2018)
5) section 7: conclusions: please add the role of microRNA in the management of headache (see J Clin Med. 2019 Jun 27;8(7):928. doi: 10.3390/jcm8070928)
Author Response
Reviewer #1
1) Methods are missing, please add the methods used to choose the manuscript.
Answer: Thank you for your suggestion. Indeed, we prepared this manuscript as a narrative review rather than a systematic one. Therefore, we do not present methods. To avoid the confusion, we revised the end of Abstract and Introduction as follows: “This narrative review …”.
2) Please add a table for each section in order to do more easy the lecture
Answer:
Following the referee’s advice, we added tables to the 3rd and 6th sections of the manuscript. For the other sections the figures presented in the manuscript best summarize the contents.
3) section 5 animal studies, this section can be deleted with the related figure
Answer:
Thank you for your comment. Translational research is an interface between basic laboratory science and its application to clinical settings. The aim of this manuscript is to explore new paths in the diagnosis and therapy of migraine and gain new insights into the pathogenesis of migraine. Inflammation plays an important role in the sensitization process that leads to enhanced responsiveness of target tissues in the CNS as well as PNS. Animal studies well support this idea, but clinical research has yet to be conducted. The purpose of this section is to summarize recent advances in the animal studies of migraine research. The concept of neurogenic neuroinflammation in the trigeminal ganglion could explain the findings of inflammatory markers in patients with migraine and may lead the way towards elucidating and treating migraine chronification. Thus, we would like the section to remain in the manuscript.
4) section 6: treatment page 12 regarding NSAIDs these drugs are commonly used in the management of headache also in chieldren so add more data regarding these drugs (see the manuscript "Headache. 2014 Feb;54(2):313-24. doi: 10.1111/head.12162." Please add also the role of PEA in the management of headache (see Front Neurol. 2018 Aug 17;9:674. doi: 10.3389/fneur.2018.00674. eCollection 2018)
Answer:
Thank you for your comment, we added these data to the modified manuscript.
5) section 7: conclusions: please add the role of microRNA in the management of headache (see J Clin Med. 2019 Jun 27;8(7):928. doi: 10.3390/jcm8070928)
Answer:
Thank you for your suggestion, we updated the manuscript.
We all appreciate your precious time and valuable comments. We sincerely hope the quality of manuscript is improved to suffice for publication.
Reviewer 2 Report
This is a well done and fairly up-to-date review
on neurogenic inflammation in migraine. My only suggestion is to review the last part relating
to new drugs because some of these are now available,
such as gepants, in example.
I also suggest checking the text as there is
some typing error
Author Response
Reviewer #2
This is a well done and fairly up-to-date review on neurogenic inflammation in migraine. My only suggestion is to review the last part relating to new drugs because some of these are now available,
such as gepants, in example.
Answer:
We appreciate your suggestion. We added the data to the manuscript.
I also suggest checking the text as there is some typing error.
Answer:
Thank you for your suggestion. We have carefully checked the full text and corrected it.
We are all grateful to your critical reading and expert opinion. Thank you.